# Tunneling Nanotubes and Tumor Microtubes in Cancer

**DOI:** 10.3390/cancers12040857

**Published:** 2020-04-01

**Authors:** Cora Roehlecke, Mirko H. H. Schmidt

**Affiliations:** 1Institute of Anatomy, Medical Faculty Carl Gustav Carus, Technische Universität Dresden School of Medicine, 01307 Dresden, Germany; mhhs@mailbox.tu-dresden.de; 2German Cancer Consortium (DKTK), Partner Site Dresden, 01307 Dresden, Germany; 3German Cancer Research Center (DKFZ), 69120 Heidelberg, Germany

**Keywords:** tunneling nanotubes (TNTs), tumor microtubes (TMs), cancer cell biology, intercellular communication, intercellular transfer, tumor microenvironment

## Abstract

Intercellular communication among cancer cells and their microenvironment is crucial to disease progression. The mechanisms by which communication occurs between distant cells in a tumor matrix remain poorly understood. In the last two decades, experimental evidence from different groups proved the existence of thin membranous tubes that interconnect cells, named tunneling nanotubes, tumor microtubes, cytonemes or membrane bridges. These highly dynamic membrane protrusions are conduits for direct cell-to-cell communication, particularly for intercellular signaling and transport of cellular cargo over long distances. Tunneling nanotubes and tumor microtubes may play an important role in the pathogenesis of cancer. They may contribute to the resistance of tumor cells against treatments such as surgery, radio- and chemotherapy. In this review, we present the current knowledge about the structure and function of tunneling nanotubes and tumor microtubes in cancer and discuss the therapeutic potential of membrane tubes in cancer treatment.

## 1. Introduction

Intercellular communication can be occur through different indirect mechanisms such as secretion of cytokines and chemokines, microvesicles or exosomes. But it may also occur directly via cell contacts like gap junctions or synapses [1,2,3,4,5,6,7,8]. These forms of intercellular communication are most effective over short distances. Over long distances, relevant receptors that can recognize and bind diffusible factors are required to effect communication between cells. Tunneling nanotubes (TNT) are another type of cell communication, as they enable cells to communicate directly with each other over longer distances. Initially TNTs has been reported in the rat pheochromocytoma cell line PC12 [9]. TNTs are long-range intercellular cytoplasmic channels for direct cell-to-cell communication that are independent of soluble factors. Uniquely, these structures allow the rapid exchange of cellular cargos between connected, non-adjacent cells, including organelles, vesicles, molecules, ions and pathogens [10,11,12,13,14,15,16].

Increasingly, these thin membrane tubes are becoming relevant for intercellular communication. They are implicated in several functions like intercellular communication during early development, cell migration, stem cell-mediated homeostasis and regeneration. Of note, TNTs may also be used by infective or pathogenic agents as routes to spread, as has been observed in cases of advancing neurodegeneration or cancer progression and metastasis [10].

TNTs were found in different organisms and tissues in multiple cell types such as macrophages [14,17], neuronal cells [12], endothelial progenitor cells [18], mesenchymal stromal cells [16,19], mesothelial cells [20,21], dendritic cells and monocytes [22,23], pigment epithelium cells [24], T-cells [25,26], B-cells [27], neutrophils [28], cardiomyocytes [29], renal proximal tubular epithelial cells [30], kidney cells [31], and in neural crest in chicken embryo [32]. Similar cellular extensions called cytonemes were also observed in non-mammalian organisms like Drosophila. These cytonemes have been described to participate in stem cell signaling [33,34,35].

TNT existence in vivo was first demonstrated by Chinnery et al. in 2008, who described their occurrence in a chimeric mouse model, between donor-derived and resident MHC class II-positive corneal stromal cells [36,37]. In tumors, the first evidence for the presence of TNTs came from studies of Lou et al. [21]. This was the first demonstration of TNTs in human primary cancer cells and intact solid tumors (resected from patients with malignant pleural mesothelioma and lung adenocarcinoma). More recently, Osswald et al. observed thin tumor cell-derived membrane tubes in an animal model of malignant brain tumors in vivo. The observed membrane tubes, termed tumor microtubes (TMs), are longer and have greater diameters in comparison to TNTs observed in vitro [38]. Recently, advanced microscopical techniques were applied to characterize membrane tubes *ex vivo* and in vivo, e.g., Rehberg et al. demonstrated membrane nanotube function in vivo using new confocal intravital imaging technologies [39].

Intercellular communication plays a major role in tissue homeostasis and is an essential factor factor for tumor development, organization and its resistance to therapeutic approaches [40,41,42], as tumors are highly heterogenous [43,44,45]. Communication between cancer cells and their microenvironment is a process that involves a variety of resident and infiltrating host cells and a diverse range of mechanisms. Non-cancerous microenvironmental stromal cells are a heterogenous group including mesenchymal stem cells and their derivatives, cancer-associated macrophages, fibroblasts, and a wide variety of immune and inflammatory cells. The tumor microenvironment contributes to tumor progression and survival of tumor cells [46,47,48]. Treatment of several malignant and invasive solid tumors, e.g., cancers of the brain, pancreas, colon and ovaries is restricted by an insufficient knowledge of intercellular communication in the tumor microenvironment [42,49].

Mounting evidence suggests that intercellular communication by TNTs and TMs may contribute to tumor survival and progression. These membrane tubes can interlink cells over considerable distances. In a solid tumor, cancer cells may be spread, so that direct communication via gap junctions, microvesicles or exosomes is improbable if not impossible. In such cases, TNTs and TMs may act as spatial communication guides, allowing direct physical contact at distance between signal-sending and signal-receiving cell membranes. They overcome the challenge of signal transport through tortuous structures within the tumor microenvironment and can provide spatial restriction, as well as specificity. In this way, TNTs and TMs support maintenance of tumor microenvironment and have been suggested to play an important role in tumor microenvironments.

## 2. Structure of TNTs

TNTs were initially characterized as F-actin containing thin membranous channels connecting two or more cells over short to long distances [50]. F-actin depolymerization drugs inhibit TNT formation [24,51,52]. Besides F-actin, microtubules or cytokeratin filaments are also detected in TNTs in a few cell lines [15,53,54]. Structurally, TNTs vary in width from 50 to 1000 nm, and in length from a few to 100 μm [21,55,56]. They do not touch the substrate [50]. TNTs can exhibit membrane continuity between connected cells by open-ended TNTs at both ends [50] or they have interposed gap junctions (close-ended TNTs) [11].

The TNTs were found to act as a cellular transport system between cells. So far, proposed functions of TNTs are long-distance exchange of different cellular components, ranging from proteins, genetic materials including microRNA and siRNA, up to other cytoplasmatic cargos like mitochondria, Golgi vesicles, and even viruses [11,18,19,21,24,29,52,53,57,58,59,60,61,62,63]. The continuity in plasma membrane and cytoplasm of connected cells allows inter-cellular transport and is mediated by cytoskeleton fibers [55]. Live cell imaging revealed that TNTs are transient structures with a lifetime of minutes to several hours [9,51,64].Two different mechanisms of *de novo* TNT formation were reported so far. In the first mechanism of TNT formation, TNTs are established by a directed outgrowth of a filopodium-like protrusion toward a neighboring cell [9]. In the second mechanism of TNT formation, TNTs are formed by dislodging of attached cells after an initial close contact [13,25,53,54]. In both cases, the process of forming tubular membrane protrusions is based on actin (reviewed in [54,65]).

A potential key factor for TNT formation is M-Sec, also known as TNFAIP2 (tumor necrosis factor α-induced protein), which interacts with the small GTPase RalA [17,66,67]. In bladder cancer cells, RalGPS2 is involved in TNT formation [68]. RalGPS2 acts independent of Ras as a guanine nucleotide exchange factor (GEF) for RalA. In HeLa cells, TNT formation required the action of the GTPase Rab8 [69]. Elevated levels of p53 were observed as essential for TNT formation by astrocytes but not in other cell types such as PC12 cells, OCI-AML3 (acute myeloid leukemia), p53-null human osteosarcoma cell line SAOS-2, and murine bone marrow-derived mesenchymal stem cells (MSC) [52,70]. This illustrates how variable signaling mechanisms for TNT formation are in different cell types or models [71]. TNT-like structures such as cytonemes have also been related to actin-based signal transduction [72,73] but unlike TNTs/TMs, they don’t establish tunneled interfaces with recipient cells. Huang et al. could demonstrate that cytoneme-mediated signaling in Drosophila development requires glutamate signaling as key factor [74].

## 3. Structure of TMs

In a model of glioblastoma in mice, Osswald et al. studied tumor development over one year using in vivo multiphoton laser scanning microscopy [38]. They detected the growth of TMs that formed in patient-derived glioblastoma cell lines xenografted into mice. Interestingly, these TMs were long-lived (days to weeks). TMs have a high content of F-actin, a key feature in most membrane nanotubes [9]. They are ensheated by a continuous cell membrane and contain myosin IIa and microtubules. They were also found to be positive for protein disulphide isomerase, partly positive for β-catenin, β-parvin, and the glial cell marker GFAP, but were found to be negative for N-cadherin, myosin X, and the neuronal cell marker β-tubulin III [38]. Branching was observed frequently in which more dynamic thin membrane tubes form from more stable, thicker ones. TMs contain microvesicles and mitochondria, suggesting that there is vesicle trafficking and also local ATP production in the membrane tubes. TMs have been shown to mediate predominantly transport of calcium and small molecules between cells. TM connections are composed of connexin 43 (Cx43) gap junctions. It was demonstrated that Cx43 expression depends on the ability of glioma cells to form TMs [38]. Interestingly, Cx43 containing gap junctions were frequently observed in TMs that connect two glioma cells, but also in TMs of different glioma cells crossing each other. Cx43 is also abundant in non-malignant astrocytes and is the most frequent connexin subtype in the CNS, particularly in glioblastoma cells [38]. Further studies of TMs are necessary to investigate if TMs are also formed in other tumor entities.

Only recently it was demonstrated that subpopulations of glioma cells form bona fide AMPA-type glutamate receptor-dependent synapses with neurons [75,76]. These glutamatergic neurogliomal synapses (NGS) between presynaptic neurons and postsynaptic glioma cells are frequently located on interconnected TMs (Figure 1) [75]. NGS are not formed in less malignant primary brain tumors, oligodendroglioma and meningioma.

TMs share a lot of structural characteristics with TNTs, such as predominant filamentous actin (F-actin), mitochondria, microvesicles or Cx43 containing gap junctions and Ca^2+^-signaling [38,77]. Differences between TNTs and TMs exist regarding their width (TMs: 1–2 μm, TNTs: < 1 μm), maximum length (TMs: > 500 μm, TNTs: about 100 μm) and lifetime (TMs: hours to days, some > 100 d, TNTs: minutes, up to 60 min).

Pontes et al. had demonstrated that glioblastoma cells are connected by a tubular network with typical features of TNTs [78]. They presume that this might reflect differences between in vitro and in vivo experiments to study TNTs. Such differences were found to be characteristic and indicated a higher structural stability in vivo compared to in vitro, which might origin in much longer time intervals for growing and connecting and hence increased functionality.

## 4. Technical Challenges Regarding Membrane Tubes

Studying the function of TNTs is challenging because their fragile nature is hard to preserve upon tissue fixation, making in vitro experiments inevitable. In addition, to date specific molecular or structural markers of TNTs and TMs are missing. Hence visualizing their morphologic features and assessing their functional feature, particularly with regard to the intercellular transport of cargos, remain as sole instruments to identify TNTs and TMs. These thin membrane tubes are difficult to be visualized in tumors, especially in predominantly stromal tumors that have a worse prognosis than non-stromal cancers [79,80,81]. TNTs and TMs can be studied in clinical tumor samples, but this requires sophisticated and reliable detection and 3D visualization methods to observe these long, thin structures. A certain overlap seems possible between TNTs and TMs due to different investigation methods, although recent data indicate that both TMs and TNTs show molecular and functional heterogeneity [15,82]. It also remains to be elucidated if they can be subclassified according to structural features. Further studies of these membrane tubes in their native microenvironment are essential for a comprehensive understanding. In vivo examination of these delicate structures remains a major barrier to the study of the function of membrane tubes in cancer. In this respect, advanced intravital microscopy techniques are expected to provide the greatest gain to enhance our knowledge about their distinct characteristics.

## 5. Occurrence of TNTs/TMs in Cancer Cells

TNTs and TMs are also present in cancer cells [50,54,56,58]. Several studies reported TNTs in cell lines of bladder carcinoma, urothelial carcinoma, breast cancer, cervix carcinoma, colon cancer, glioblastoma, leukemia, mesothelioma, osteosarcoma, ovarian cancer, adenocarcinoma, pheochromocytoma, prostate cancer, and squamous cell carcinoma (Table 1). In addition, TNTs were also detected in different tumor types explanted from patients including mesothelioma [21], ovarian cancer [57,58], osteosarcoma [83], pancreatic cancer [84], squamous cell carcinoma [85], and laryngeal carcinomas [86]. In recent years, understanding of membrane tubes between cancer cells expanded considerably by the reports of TMs, which were followed up in human gliomas implanted in the mouse brain for up to one year [38,75,82,87]. Cancer cells can use TNTs/TMs to communicate both among malignant cells (homotypic interactions) and cells of the tumor microenvironment (heterotypic interactions). Membrane tube-connected cells of tumor microenvironment are stromal cells like endothelial cells, pericytes, mesenchymal stem cells, neurons, fibroblasts, osteoblasts, macrophages (Table 1).

The appearance of intercellular membrane tubes may vary and depends on the tumor type. It was demonstrated that epithelial-to-mesenchymal transition (EMT) stimulates TNT formation in mesothelioma [21]. Accumulating evidence suggests that cancer cells with lower proliferation rate that have a higher lipid raft content generate more TNTs [56], e.g. it could be shown that malignant mesothelioma cells generate more TNTs than benign mesothelial cells [56].

Investigation of resected human tumor samples revealed that about 63% of malignant astrocytomas formed intercellular TMs. In contrast, TMs were observed only rare in oligodendroglioma cells of human tumor specimens or in patient-derived oligodendroglioma cells implanted in mice [38]. By live-cell STED nanoscopy it was shown that membrane HSP-70 supports the formation of cell-to-cell connections via TNTs in glioblastoma and also in mammary carcinoma cells [88].

Solid tumors such as gliomas exhibit large intratumoral and intertumoral cellular heterogeneity. Subpopulations, even very small one, assume specialized functions essential for cancer progression [45,89]. Intercellular communication between these subpopulations may contribute to heterogeneity in the tumor itself and also in its microenvironment by TNTs and TMs. It was observed that glioma cells show a pronounced heterogeneity regarding TM formation. Interestingly, only a subpopulation of glioma cells formed TMs and only few of these TM-positive cells connected with other tumor cells with TMs [38]. Jung et al. observed glioma cell populations with divergent TM subtypes, of which one particularly invasive glioma cell subpopulation displayed 1-2 TMs and was regulated by the membrane protein tweety-homolog 1 (Ttyh1). Another, less invasive subpopulation formed multiple TMs that created a tight multicellular network within the glioma [82]. Furthermore, such invasive TMs share some morphological features with other cellular protrusions, such as invadopodia [90] or filopodia [91].

## 6. Functions of TNTs and TMs in Cancer

### 6.1. Cancer Growth and Invasion: Homotypic Interactions between Cancer Cells

TNTs and TMs have been associated with cancer cell growth and invasion. Cell migration at developmental stages requires guided growth of extending cellular protrusions [110], which appears to be also the case for neuronal cell migration in the developing brain [111]. When comparing TMs with developing neuronal growth cone tips in vivo, Osswald et al. observed a similar morphology with frequent dendritic branching [38]. TMs showed morphological features of axonal growth cones at the leading edge of invasion in vivo, suggesting that similar pathways might be responsible for TM formation. Similar to neuronal growth cones located at the tip of the growing axon during development, the tips of membrane tubes are highly dynamic [38,112]. The number of TMs increases with tumor progression. They are used as guiding rails for dissemination cell nuclei after karyokinesis, indicating that TMs are important for tumor cell spreading (Figure 2).

Imaging TM formation in gliomas in vivo revealed a multicellular network of glioma cells connected by TMs [38]: Intercellular TMs can form consequently to cell division to ensure persistent contact between daughter cells, but also between non-related glioma cells. Therefore single cancer cells are interconnected by membrane tubes to represent one large syncytium. Notably, the formation of such syncytium correlates with the prognosis of brain tumor malignancy [38]. It has been reported that particularly, TMs of glioma cell subpopulation with 1–2 TMs infiltrated the brain at the invasive front. After this invasive glioma cell subpopulation has infiltrated brain tissue, the cells settle and transform into cells interconnected by multiple TMs, as indicated by the expansion of this subpopulation during tumor progression [38]. Consistently extensive and long-range intercellular calcium waves largely restricted to astrocytoma cells were observed in various tumor regions [38]. In addition, a subgroup of astrocytoma cells was demonstrated to explore the perivascular niche which is associated with the cancer stem cell pool and where the cells remained for months [113]. The perivascular space is one of the two routes mainly used by glioma cells to invade brain tissue, the other one being the surrounding area of axons [114,115]. About 20% of glioma TMs is following axons in the brain suggesting that TM-derived migration of glioma cells may occur along neuronal axons as guiding cues for tumor cell dissemination in gliomas.

Important to note, Cx43 gap junction-mediated communication stabilizes TMs [38]. In consequence, Cx43 deficiency was observed to lead to a reduction in tumor size. Ttyh1, which is found during neuronal development [116] and is involved in neurite outgrowth [117], was demonstrated to regulate TM morphology and function [82]. Data point out that Ttyh1 is important for TM-dependent invasion and proliferation of astrocytoma cells, resulting in tumor progression in the brain. Consequentially, Ttyh1 downregulation in glioma cells caused reduced tumor progression and prolonged survival of mice. The chromosomal location of the ttyh1 gene to chromosome 19q is interesting, since codeletion of 1p and 19q is a characteristic in oligodendrogliomas, a brain tumor subtype with a better prognosis as compared to astrocytoma or glioma [118].

During neurogenesis, the neuronal growth associated protein-43 (GAP-43) is highly expressed in axonal growth cones and appears to be required for TM formation between glioma cells [38,119,120]. Further, it is induced by neurotrophin receptor signaling and is required for neuronal progenitor cell migration [121,122,123]. Ectopic GAP-43 expression in non-neuronal cells induced a large amount of long, fine filopodial processes extending from the periphery. Likewise, neuronal cells exhibited more and longer processes [124]. Remarkably, lesion-induced nerve sprouting and terminal branching during reinnervation were observed to be more efficient in a GAP-43-overexpressing mouse model [125]. Concerning TNT/TM, it can be concluded that GAP-43 is engaged by glioma cells to form and elongate TMs and to propagate a functional TM network by driving cell migration [38]. It is highly expressed in the tips of growing TMs that resemble growth cones but are negative for nestin. Consistently, the structure and branching of TMs were affected in primary glioblastoma cell lines with a genetic knockdown of GAP-43. This resulted in a decreased proliferation capacity and invasion speed and hence an impaired spreading of glioma cells. Consistently, elevated levels of GAP-43 were found in TM-rich human glioma cells and primary stem-like cell lines that had no codeletion of 1p/19q compared to oligodendroglioma cells with a codeletion of 1p/19q [38].

Similarly, TNTs seem to be leading structures for mesothelioma cell invasion. Lou et al. could show that TNTs form at the invasive edge in mesothelioma cells in vitro by using modified wound-healing assays [21]. As revealed by videomicroscopic imaging, TNTs were regularly formed during proliferation and migration of mesothelioma cells to fill intercellular gaps in the developing tumor tissue.

### 6.2. Cancer Growth and Invasion: Heterotypic Interactions

Mounting evidence suggests that intercellular communication between cells of the tumor and the surrounding stroma within the tumor microenvironment is essential for invasive cancer types to progress [126]. Hanna et al. demonstrated that heterotypic TNTs between breast tumor cells and macrophages were responsible for induction of an invasive tumor cell morphology in dependence on signaling via the EGF-EGFR (epidermal growth factor-epidermal growth factor receptor) pathway [94]. In this study, authors demonstrated that tumor cells preferentially migrated alongside macrophages towards the endothelium, and that tumor cell invasion occurred at elevated levels in vitro and in vivo.

Neurogliomal synapses (NGS) on TMs foster glioblastoma progression by promoting tumor growth [75]. NGS were found to strongly contribute to the proliferation promoting effect of glutamate on glioblastoma cells and consequently stimulate tumor growth [75]. Therefore, gliomas integrate into electrical networks and depolarizing current promotes tumor progression. The glioblastoma cell subpopulation functionally linked to neurons in vivo showed a significantly higher invasive potential than other glioblastoma cells [75]. Interestingly, communication between astrocytes and neurons plays also a role during development of neural tissues. Based on studies on the growth of the hippocampus, Wang et al. suggested that neurons use TNTs during their maturation to arrange electrical coupling and for exchanging calcium signals with astrocytes [59].

### 6.3. Tumor Cell Survival

Osswald et al. (2015) observed a self-repairing mechanism inside of TM-connected BTPCs (brain tumor propagating cells) after photon damage by lethal laser irradiation [38]. Upon ablation of a single TM-connected glioma cell, the dead cell was approached by new TMs that extended from surrounding live glioma cells. A new cell nucleus translocated to the ablated cell using TMs within few days. In contrast, this repair mechanism was only rarely observed in ablated non-TM-connected cells. Photon damage of more cells led to rapid expansion of glia cell TMs to the damaged area with a subsequent local enhancement in tumor cell density. The fact that stress conditions promote membrane tube formation indicates that TMs represent some kind of survival mechanism. Consistently, hypoxia present in various kinds of cancer can also foster TNT formation. A hypoxic condition is a hallmark in many malignant solid tumors, e.g., hypoxia elevates the formation of TNTs in cancer of the ovaries or colon [64,108]. The highest effect was seen in chemoresistant ovarian cancer cells and notably could not be suppressed by applying a compound that otherwise suppressed TNTs in sensitive cells. Wang and Gerdes found that PC 12 cells treated with UV light were rescued when cocultured with untreated cells due to TNT-mediated mitochondria transfer along microtubules in connecting TNTs [15] (Figure 2).

Communication between tumor and stromal cells is crucial for tumor survival. Cancer cells may influence the microenvironmental cells to promote pathways that support cancer cells. Besides receiving mitochondria, cancer cells can also employ signal transfer to modify their microenvironment, thus favoring tumor progression. Polak reported a signal from leukemia cells of an acute lymphoblastoma to stromal cells of the bone marrow (MSCs) through TNTs with consequential release of survival promoting cytokines [103]. It was also observed that activated MSCs translocate mitochondria to leukemia cells of acute lymphoblastic leukemia (ALL) during oxidative stress [104]. Marlein et al. reported that MSCs translocated mitochondria to acute myeloid leukemia (AML) blastocytes by AML-derived TNTs [105]. They show that NOX2-derived superoxides generated from AML blasts cause mitochondrial transfer.

### 6.4. Modulating of Tumor Cells and Environment via TNT/TM Mediated Cargo

#### 6.4.1. Transfer of Mitochondria

Mitochondria can be transferred between cells by TNTs [24,50]. Such transfer of mitochondria through TNTs was observed between the tumor microenvironmental cells and cancer cells and between cancer cells in many different cancer types, including mesothelioma [21], leukemia [105,127,128], squamous cell carcinoma [85,86], breast [58,93,129], bladder [92], ovarian cancers [58,108] as well as in glioblastoma [130,131]. The metabolic effects of the acquired mitochondria were demonstrated in numerous in vitro studies. Of note, TNT-mediated mitochondrial translocation modified energetic metabolism of the receiving cell including elevated OXPHOS and ATP production in the various cell systems studied [93,127,132,133]. The acquisition of mitochondria by cancer cells resulted in the increase of proliferation rate and invasiveness. Caicedo et al. demonstrated that MSC mitochondria were able to increase proliferative and invasive capacity in a breast cancer cell line [93]. Lu et al. observed that increased invasive behaviour of bladder cancer cells succeeded TNT mediated trafficking of mitochondria between highly invasive bladder cancer cells and less invasive urothelial cells [92].

#### 6.4.2. Transfer of microRNA

It has also been reported that TNTs facilitate transport of oncogenic microRNAs (miRNA) between tumor cells and between tumor and surrounding stromal cells (Figure 2) [57,83]. Transfer of miRNA was shown to occur between the tumor and the surrounding stromal cells of tumor microenvironment, e.g. between osteosarcoma cells and stromal osteoblasts or between ovarian cancer cells and cells of ovarian epithelium [57]. Furthermore, horizontal miRNA transfer between metastatic cancer cells and endothelial cell by TNTs has recently been linked to metastatic invasiveness [95]. It was observed that metastatic tumor cells predominantly form TNT-like “nanoscale intercellular membrane bridges” with endothelial cells [95]. Connor et al. assumed that such tubes foster communication between cancer cells and healthy endothelial cells, thereby transforming a healthy endothelium into a pathological endothelium. In this study it was demonstrated that breast cancer cells form TNT-like structures for transferring miRNA from the tumor to the endothelium and induce a pathological phenotype. Such miRNAs were shown to function as signaling regulators of tumor cell migration and invasion [134,135].

#### 6.4.3. Transfer of Oncogenes

Recently it was shown that transfer of the RAS oncogene KRAS between cells can be mediated by TNTs of colon cancer cells [97]. Such intercellular translocation of oncogenes results in a heterogeneous distribution of mutant KRAS in cells with endogenous wild-type KRAS. KRAS, one of the most frequently mutated oncogenes in cancer, is known to act as a critical driving force in cancers like colorectal and pancreatic cancers [136,137,138]. Oncogenic KRAS signaling modulates the tumor environment profoundly, including responses of the immune system towards the tumor, cancer-associated fibroblasts, and angiogenetic responses [139]. Hence amplifying oncogenic KRAS spreading via TNT mediated translocation may have important consequences for tumor progression, implying that TNTs have the potential to reprogram malignant cells and alter cells in the tumor environment.

#### 6.4.4. Transfer of Exosomes and Other Cargoes

TNTs and exosomes may work synergistically. In human mesothelioma cells it was demonstrated that exogenous tumor exosomes induced enhanced TNT formation of TNTs as additional intercellular transport tracks [83]. As shown by electron microscopy, exosomes accumulated at the base of TNTs and also extracellularly, suggesting that exosomes may act as a potent chemotactic stimulus for TNT formation and guidance (Figure 2). Burtey et al. characterized a route of intercellular trafficking of transferrin receptor tagged with mCherry between cancer cells by a mechanism depending on Rab8 and intercellular contact [69]. Transfer occurred between several types of cancer cells, and comparatively less transfer occurred between non cancer cells, namely normal rat kidney epithelial cells.

### 6.5. Angiogenesis

Angiogenesis is a hallmark of cancers. Angiogenic switch is a process through which tumor cells develop an angiogenic phenotype, thus initiating angiogenesis [140,141,142,143]. The study of Errede et al. implicates a primary role of TNTs in tumor angiogenesis [102]. It was hypothesized that TNTs may promote angiogenesis through establishing contacts between pericytes but also between pericytes and endothelial cells (Figure 2). It was found that pericytes form TNTs, pointing out to pericytes as the primary source of TNTs to initialize and foster growth and branching of vessels in glioblastoma as well as in the developing cerebral cortex. Ultra-long TM-like TNTs were observed to connect the walls of distant vessels and short TNTs to connect juxtaposed sprouting vessels with each other or a vessel sprouting with its facing vessel. TNTs that connected human microvascular endothelial cells were also shown to contain lipid droplets. These lipid droplets were able to increase in number after VEGF treatment, followed by the activation of certain signaling [144]. This leads to increased cell motility and additional TNT formation.

### 6.6. Treatment Resistance

For invasive tumors like glioblastoma, the current standard care of therapy consists of a combination of surgery, radiation, and chemotherapy but the therapeutic success remains poor [145]. This is due to the fact that TNTs and TMs contribute to treatment resistance by establishing networks between tumor cells. Such networks can extend to tissue surrounding the tumor, thereby increasing its malignant potential.

#### 6.6.1. Resistance to Surgical Lesions

TM-connected glioblastoma network was able to repair itself when a laser surgical excision was performed. In a model of intracranially implanted patient-derived BTPCs in mice, Weil et al. demonstrated, that glioblastoma cells formed and extended more TMs toward the lesioned area [87]. This study demonstrated that a surgical lesion is able to trigger excessive repopulation of the lesion with glioblastoma cells mediated by TMs [87]. This is supported by the finding that the tumor cell density significantly exceeded unlesioned parts of the brain. Consistently, it has been shown that the majority of tumors, though colonizing the resection margin, reappear at the resection edge. Moreover, genetic targeting of GAP-43 could suppress tumor growth after surgery by impairing formation and function of TMs and therefore could reduce tumor recurrence upon surgery by repopulation.

#### 6.6.2. Resistance to Chemotherapy

It was observed that chemotherapeutic agents have the potential to affect TNT formation and cargo trafficking. TNTs/TMs have been shown to mediate chemoresistance for example in ovarian and breast cancer cells [58,108], in pancreatic cancer cells [84], leukemia cells [104,106,127] and in glioblastoma cells [87]. Desir et al. demonstrated that chemoresistant ovarian cancer cells formed more TNTs than chemosensitive cells when oxidative stress was subjected to the cells [108]. Pasquier et al. observed mitochondrial translocation from endothelial cells to tumor cells via TNTs and assume that chemoresistance was associated with such mitochondrial trafficking [58]. This finding may be supported by reports on MCF7 breast cancer cells that displayed elevated doxorubicin resistance after receiving mitochondria from endothelial cells. Similar results were observed for hematopoietic malignancies. Wang et al. showed that transfer of mitochondria supported mesenchymal stem cell-induced chemoresistance in T cell acute lymphoblastic leukemia cells due to reduced ROS levels [106]. Moschoi et al. demonstrated that mitochondrial transfer from bone marrow-derived stromal cells can protect acute myeloid leukemic (AML) cells when exposed to chemotherapy [127]. In this study, cytarabine (ARA) increased communication and trafficking of AML cells with bone-marrow-derived MSCs by incorporation of mitochondria in AML cells. This effect was also present when AML cells were treated with the topoisomerase II inhibitor etoposide and the anthracycline doxorubicin.

Similarly, MSCs activated by chemotherapy were able to prevent apoptosis in acute lymphoblastic leukemic (ALL) cells upon therapy by using TNTs to transfer mitochondria [104]. Consequently, reduced mitochondrial trafficking prevented this rescue effect.

Like TNTs, TMs contribute to chemotherapy resistance as well, as seen in tumor cells linked to each other by TMs and in tumors with abundant TMs that showed increased resistance against temozolamide treatment [87]. The mechanisms of this survival benefit after chemotherapy of glioblastoma cells connected by numerous TMs compared to tumor cells that were not linked to each other by TMs are not entirely clear. Intercellular dissemination of regulatory factors by membrane tubes could be considered as a key player in developing resistance to chemotherapy. The authors assumed that the tumor cell network may foster broad dissemination of chemotherapeutic agents, thereby preventing temporary critical high drug concentrations. This is supported by the fact that TMs of connecting glioma cells and crossing intercellular TMs contain numerous Cx43 gap junctions [38]. Such gap junctional communication between astrocytes and glioma cells may protect the tumor from chemotherapy, too [146,147]. Therefore it is also possible that non-malignant cells establish gap junction connections with glioblastoma cells to participate in this protective network. In conclusion, the functional syncytium that is established by abundant TM connections between glioma cells and tumor microenvironment provides protection against chemotherapy.

#### 6.6.3. Resistance to Radiotherapy

Glioma cells connected by TMs were also shown to be more robust and display higher survival rates under standard radiotherapy, presumably because the TM network promotes a more stable multicellular homeostasis [38]. TM-connected cells increased their TM number and thereby communicative calcium spreading. Consistently, knockdown of Cx43 impaired the radioprotective effect of the TM network. Interestingly, only the TM-devoid and unconnected glioma cells died in relevant numbers after radiotherapy. Radiotherapy was able to induce cell death in TM-devoid and hence unconnected glioma cells by disturbing the intracellular free calcium equilibrium compared to non-irradiated and TM-connected cells that sustained homogenous intracellular free calcium levels during irradiation. Another study demonstrated that an elevation of intracellular free calcium is a key event in the induction the execution phase of apoptosis [148]. Osswald et al. suggested the intercellular TM network to serve as a calcium distributor that prevents critical elevations of free intracellular calcium in single tumor cell to sustain nonlethal levels, thereby preventing cytotoxic effects of irradiation and subsequent apoptotis in glioma cells [38]. In the review of Matejka and Reindl they focused especially on the role of TNTs in the radiation response of cells upon radiation treatment using X-rays and alpha-particle treatment [65].

## 7. Conclusions and Perspectives

TNTs and TMs came into focus as novel intercellular communication mediators. *In vitro* data revealed their potential to the onset and progression of tumors by forming communication networks within the tumor microenvironment that increase malignancy by allowing for distribution of metastatic potential. Moreover, recent findings revealed that glioma cells are capable of forming TNTs/TMs in vivo in correlation with the tumor’s metastatic potential and demonstrated that neurons can form synaptic contacts with TNTs/TMs from tumor cells. Pharmacological perturbation of TNTs/TMs consequentially reduced the metastatic potential of tumors in vivo in experimental metastasis models. Therefore disruption of TNTs/TMs connections has the potential to decrease resistance and recurrence rates of cancer by pharmacologically disconnecting tumor networks. On the other hand, TNTs/TMs can also provide an effective network for long-range cellular therapeutic drug delivery following oncolytic viral treatment. Furthermore, TNTs/TMs mediated the bystander effect after oncolytic viral infection and administration of nucleoside analogs [107]. Thus, TNTs/TMs offer potential cellular conduits for drug delivery and hence amplification of cancer-targeting drug effects. While first studies of TNTs/TMs were performed in vitro, more sophisticated study designs involving ex vivo tumors and finally in vivo studies in animal models emerged. Although knowledge about TNTs/TMs increased over the past years, many complex and intriguing aspects remain to be elucidated. For example, identifying molecular markers for TNTs/TMs in vivo remains a major challenge for understanding TNT/TM function. Exploring how TNTs/TMs connect, how they support intercellular communication, or how they are temporally and spatially regulated, has emerged as an essential issue for understanding intercellular communication and interaction between cancer cells and their microenvironment. Exploring these aspects will provide key insights into the regulation of cancer progression.

## Figures and Tables

**Figure 1 cancers-12-00857-f001:**
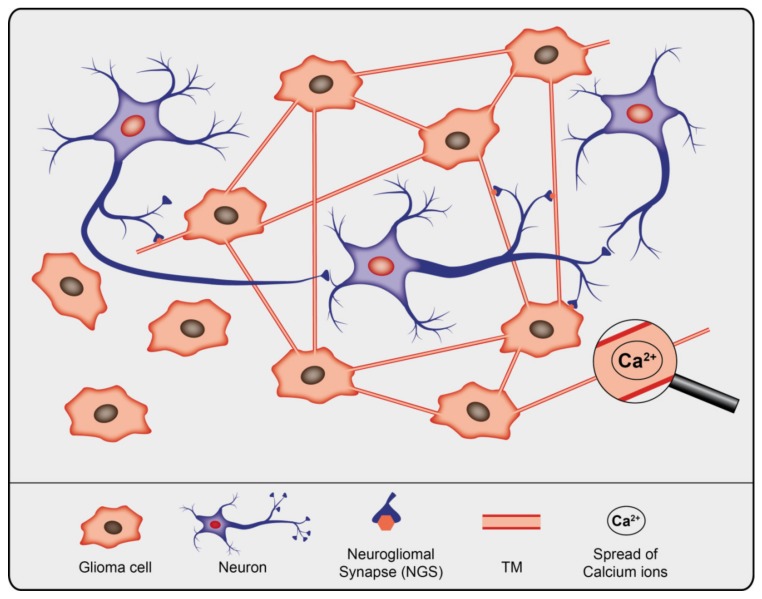
Schematic illustration of gliomal network, the neuronal network, and their interconnectivity.

**Figure 2 cancers-12-00857-f002:**
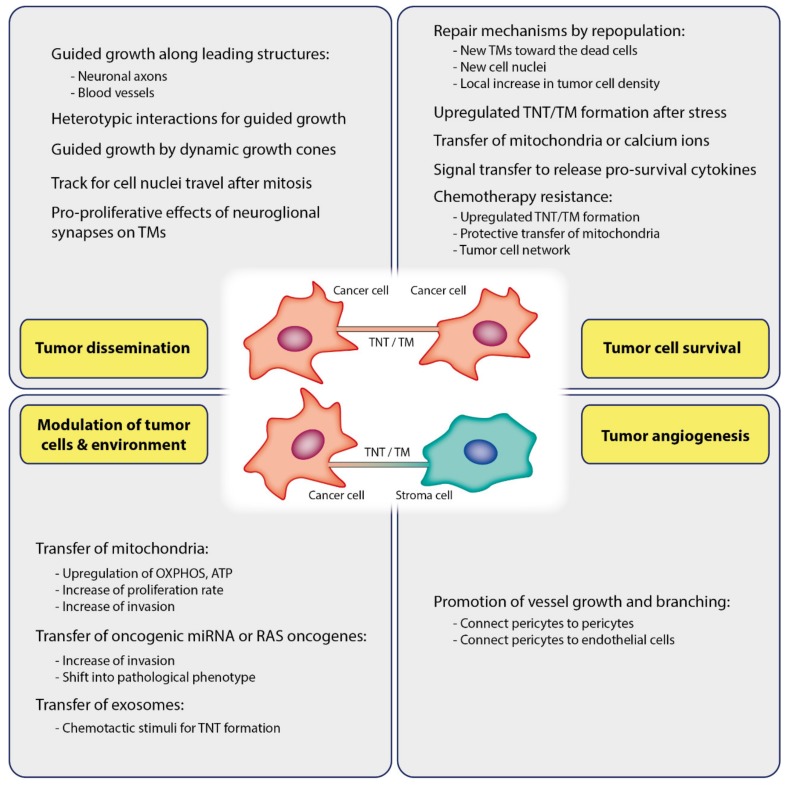
Functions of TNTs/TMs in cancer. TNTs/TMs are critically involved in tumor initiation, growth, progression, metastasis, and chemotherapy resistance.

**Table 1 cancers-12-00857-t001:** Overview of the occurrence of TNTs/TMs in cancer cell types.

	Cancer Cells or Cancer-Derived Cell Lines	Cancer-Related Microenvironmental Cells	Cancer Model In Vivo	TNT/TM: Interaction	Ref.
Bladder cancer	Human bladder carcinoma cell line 5637 (HTB-9)			CC–CC	[68]
	Human urothelial carcinoma cell line: T24		Nude mice: xenograft model (tumor growth)	CC–TC	[92]
	Non-malignant urinary papillary urothelial tumor cell line: RT4				
Breast cancer	Human breast cancer cell line: MDA-MB231, MCF7	Human MSCs from bone marrow aspirates of healthy donors		CC–CC	[58,93]
				CC–EC	
		Human endothelial cells: E4ORF1-positive cells		CC–MSC	
	Rat mammary adenocarcinoma MTLN3 cells	Murine RAW/LR 5 monocytes/macrophages		CC–macrophages	[94]
	Mouse mammary carcinoma cells (4T1)			CC-CC	[88]
	MDA-MB231, MDA-MB435, MCF7, MDA-MB468, SKBR3	Human endothelial cells (HUVEC, primary dermal microvascular blood endothelial cells, primary dermal microvascular lymph endothelial cells)		CC-CC	[95]
				CC-EC	
		Human endothelial cell line: HMEC cells		(TNT-like structures: nanoscale intercellular membrane bridges)	
Cervix cancer	Human cervix carcinoma derived cell line: HeLa	Human NRK fibroblasts		CC–CC	[69,96]
				CC–SC	
Colon carcinoma	Human colon cancer cell lines: LoVo, HCT116, SW480, HT29, AAC1			CC–CC	[97]
	Human colon carcinoma cell line SW620			CC–CC	[98]
Glioma	Human primary glioblastoma cell lines: patient-derived from resected glioblastomas (BTPCs): S24, T269, T325, T1, P3, T1, T269, WJ, BG5, E2	Rat primary cultures of cortical neurons	NMRI nude mice and NSG mice (patient-derived xenograft models)	*In vivo* and in vitro: CC–CC	[38,75,82,87]
	Human oligodendroglioma cell lines (BT088 and BT054)		Human tumor material from patients with glioblastoma	CC-N	
	Rat C6 glioma cells	Rat primary astrocytes		CC–SC	[99]
	Human glioblastoma cell line: U87MG, U87			CC–CC	[78,88,100]
	Human glioma cell line U251			CC–CC	[101]
	Mouse glioma cell line GL261			CC–CC	[88]
		Human primary pericytes (HBVP)	Human fetal cerebral cortex, human glioblastoma tissue samples from primary tumors	SC–SC	[102]
Laryngeal cancer	Human primary culture of LSCC cells		Human LSCC tissue samples	CC–CC (CC–other cell types: ND)	[86]
Leukemia	Human BCP-ALL cell line: NALM6 and REH	Primary MSCs: human bone marrow aspirates from patients with BCP-ALL		CC–MSC	[103]
	Human BCP-ALL cell line: REH, SD1, SEM, TOM1	Human MSC cell line HS27a, Murine MSC MS5, Primary human MSC (from MNC)	Murine NSG model of disseminated SEM-derived ALL	CC–MSC	[104]
				CC–CAF	
	Human primary AML blasts (from patient bone marrow)	BMSC: CD34-positive hematopoietic stem cells from human donor)	In vivo xenograft model (NSG mice)	CC–CC	[105]
				CC–BMSC	
	Human T-ALL cell line Jurkat	Primary MSCs: bone marrow aspirates from human donor		CC–MSC	[106]
	Human primary T-ALL cells				
Lung cancer			Lung adenocarcinoma cells from patients with pleural effusions	CC–CC	[21]
			Tumor specimens from patients with poorly differentiated lung adenocarcinoma		
Mesothelioma	Human mesothelioma cell lines: MSTO-211H (CRL-2081), JMN, VAMT, H2052, H-Meso		Pleural mesothelioma cells from patients with pleural effusions	CC–CC	[21,56,83,107]
	Compared with: human immortalized normal (benign) mesothelium cell lines: LP9, Met5A		Tumor specimens from patients with malignant pleural mesothelioma and poorly differentiated lung adenocarcinoma	CC–TC	
Osteo-sarcoma	Murine osteosarcoma cell line: K7M2 cells	Murine osteoblast cell line: MC3T3		CC–CC	[57]
				CC–SC	
Ovarian cancer	Human ovarian cancer cell lines: SKOV3 (HTB-77), OVCAR3 (HTB-161)	Human MSCs	Human ovarian cancer explant	CC–CC	[58]
				CC–EC	
		Human endothelial cells: E4ORF1-positive cells		CC–MSC	
	Human ovarian carcinoma cell line: A2780, C200	Human immortalized cell line from normal ovarian epithelial cells: IOSE		CC–CC	[57,84,108]
	Human ovarian adenocarcinoma cell line: SKOV3			CC–SC	
Pancreatic cancer	Human pancreatic adenocarcinoma cell lines: MIA PaCa, S2013, CAPAN-1, CAPAN-2	Human pancreatic duct epithelial cell line: HPDE cells	Tumor pieces from patients with pancreatic cancer	CC–CC	[84]
				CC–SC	
Pheochromocytoma	Rat pheochromocytoma cells (PC12 cells)			CC–CC	[15,50,51]
Prostate cancer	Human prostate cancer cell lines: PC-3 and 22Rv1 cells	Human fetal osteoblastic cell line: hFOB		CC–CC	[109]
	Human androgen-sensitive prostate adenocarcinoma cells: LNCaP cells			CC–OB	
Squamous cell carcinoma (SCC)	Human SCC-derived cell lines: SCC2, 38, 40, 42B	Human CAF	Tumor pieces from patients with SCC	CC–CC	[85]
				CC–CAF	

**Abbreviations:** BCP-ALL: B-cell precursor-acute lymphoblastic leukemia cells; BMSC: bone marrow stromal cells; BTPC: brain tumor propagating cells; CAF: cancer associated fibroblasts; CC: cancer cells; EC: endothelial cells; HBVP: human brain vascular pericytes; HUVEC: human umbilical vein endothelial cells; LSCC: laryngeal squamous cell carcinoma; MNC: mononuclear cells; MSC: mesenchymal stem cells; N: neurons; ND: not defined; OB: osteoblasts; Ref: references; SC: stroma cells; SCC: squamous cell carcinoma; TC: (benign) tumor cells; TMs: tumor microtubes; TNTs: tunneling nanotubes.

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
