# Peer review of "Tunneling Nanotubes and Tumor Microtubes in Cancer"

_cancers, 2020, doi:10.3390/cancers12040857_

Round 1

Reviewer 1 Report

In this manuscript the authors analyzed the structure and the functions of the tunneling nanotubes and tumor microtubes in cancer.

Although, in the last few years, other works were carried out about this subject, this is an useful and interesting topic that makes the manuscript worthy of consideration. This investigation could provide a significant contribution to knowledge in the field, favoring, in future, the development of potential therapeutic approaches for cancer patients. The article should be of potential interest to a broad readership interested in basic and translational cancer research.

The paper is readily intelligible because it is well-constructed, clear and well described with figures and tables appropriate to the subject matter. The scientific background and aims are clearly explained. The conclusions logically follow from data present in the literature.

The authors have fulfilled all previous requests performed by Reviewer. For all these reasons, the paper could be considered adequate for the standards required for the publication in Cancers.

Reviewer 2 Report

The authors have well addressed my comments, and I do not have further comment regarding this manuscript.

Reviewer 3 Report

The authors fully covered my critisism.

This manuscript is a resubmission of an earlier submission. The following is a list of the peer review reports and author responses from that submission.

Round 1

Reviewer 1 Report

Reviewer have come up with a very strong concern: portions of this manuscript may be taken from other manuscripts and/or book chapters.   Otherwise the top of the 2nd page does state a strong inaccuracy- that was the first to detect TNTs/nanotubes in tumors ex vivo

Reviewer 2 Report

The authors presented the common knowledge on the connection of cell communication and cancer with regard to membrane structures such as tunneling nanotubes and tumor microtubes.

The review is well written and sound. English writing and structure should be checked such as in 2. Structure of TNTs " "... from one cell to another, which . " There is something missing.

The authors should (Matejka et al. "Perspectives of cellular communication through tunneling nanotubes in cancer cells and the connection to radiation effects." Radiation Oncology 14.1 (2019): 1-11.) cite and comment on differences and similarities. And also a citation is missing in the part of Gliomas and Glioblastoma (Reindl, et al. "Membrane Hsp70-supported cell-to-cell connections via tunneling nanotubes revealed by live-cell STED nanoscopy." Cell Stress and Chaperones 24.1 (2019): 213-221.).

Reviewer 3 Report

In this review manuscript, the authors summarize the current knowledge of tunneling nanotubes and tumor microtubes. This review is well structured with both basic concepts and recent research advance.  It is very comprehensive, including the molecular composition, structural features, functional implication of TNT and TM in tumor progression. This review merits the publication on Cancers after correcting a few minor errors.

components from one cell to another, which . This trafficking relies on cytoskeleton fibers

‘cytoplasmatic cargo’     -----  ‘cytoplasmatic cargos’ 

‘the intercellular transport of cargo’  ----- ‘ the intercellular transport of cargos’ 

neurons, fibroblasts, osteoblasts, macrophages) (Tab.1).

‘In development invasive cell migration has been shown to’ ---- not clear here

‘nuclei after mitosis indicating that’  -----‘nuclei after mitosis, indicating that’ 

Reviewer 4 Report

The manuscript tilted "Tunneling Nanotubes and Tumor Microtubes in Cancer", is in my opinion a very well written and up to date review focusing on the role of novel intercellular communication mediators as TNTs and TMs and their potential to the onset and progression of tumors. Moreover, it well resumes the state of art through a careful review of the literature in the field. However, before publication, the manuscript needs a careful reading of the text for the presence of many typos and language errors.